# The Causal Pathway of Rural Human Settlement, Livelihood Capital, and Agricultural Land Transfer Decision-Making: Is It Regional Consistency?

**Weiwen Wang [1]** , **Jian Gong [2],**** , **Ying Wang [2]** and **Yang Shen [2]**

1   College of Geography and Environmental Science, Henan University, Kaifeng 475004, China; wwwang@henu.edu.cn
2   School of Public Administration, China University of Geosciences, Wuhan 430074, China; yingwang0610@cug.edu.cn (Y.W.); shenyang@cug.edu.cn (Y.S.)
*   Correspondence: gongjian@cug.edu.cn

**Abstract:** Despite the increasing interest in understanding the mechanism of household livelihood decisions to increase household livelihood welfare, the combined role of livelihood capitals and human settlements in livelihood decisions is unclear. Therefore, in this paper we carried out extensive empirical research to explore the causal pathway between human settlements (including infrastructure, public services, and social governance) and livelihood capitals (including human, natural, physical, financial, and social capitals) on agricultural land transfer, taking employment choices as an intermediary factor. On this basis, this study analyzed the regional differences in the decision-making mechanisms of agricultural land transfer behaviors in eastern, central, and western regions of China through a multi-group structural equation model. The results demonstrated that capital accumulation can directly increase the possibility of agricultural land inflow (β = 0.130, $p < 0.01$), but can indirectly reduce the dependence on agricultural land by stimulating non-agricultural employment (β = −0.613, $p < 0.01$). The improvement in human settlement promotes the agricultural land inflow (outside the western region) and indirectly enhances the willingness to enter into agriculture. The employment choices play a significant mediating role by strengthening the livelihood capitals and weakening human settlements. To achieve the intense agricultural development and sustainable development of rural areas, the improvement of both rural human settlements and household livelihood capitals should be considered.

**Keywords:** agricultural land transfer; rural human settlement; household livelihood capital; employment choices; regional differences



## 1. Introduction

Faced with the unfair positioning of rural values and the long-term isolation of urban–rural relations in China, migration is often the main employment choice for the young generations [1–3]. In 2019, nearly 170 million peasant workers left villages and settled down in cities [4]. This transfer of the agricultural labor force has triggered significant changes in resource allocation, rural land use, and labor relations [5]. Due to this massive migration, two million hectares of agriculture land fall out of production each year in China [6]. As the material basis for human social and economic activities, land resources play a crucial role by providing the space to support various rural industrial development demands [7,8]. To give full play to the production capacity of the rural industry, it is necessary to achieve mechanized-scale operation through agricultural land transfer [9,10]. Agricultural land transfer refers to a process of the reallocation and optimization of agricultural land among different management bodies [11], which aims to facilitate the transfer of surplus rural labor and improve the efficiency of land use [12].

Rural households, as the most important actors in rural areas, are the decision-makers in agricultural land transfer [13]. The sustainable livelihood framework (SLF) is widely used to understand how rural households make livelihood decisions to seek more profitable and stable livelihood strategies when faced with changes to their livelihood capitals (the resources available to households for their livelihood and development), the external environment, policies, public resources, and other conditions [14,15]. The SLF emphasizes the role of livelihood capitals in the maintenance of sustainable livelihoods [16,17]. Exploring how households form livelihood strategies and make land use and employment choices based on their livelihood capitals can enhance the understanding of large-scale intensive land use and provide insights to improve household livelihoods. Much of the current literature on agricultural land use decisions has centered on food production, water and fertilizer management, crop choices, and agricultural inputs and outputs [18–22]. Little attention has been paid to how livelihood capitals affect agricultural land transfer, and how they affect agricultural land transfer through rural employment choices.

In addition to income growth, another important motivation for rural households related to migration is to enjoy high-quality human settlements in cities [23]. Households have the tightest connections with rural human settlements, which are the sum of all facilities and services supporting household production and living [24]. Households shape rural human settlements, and the characteristics of human settlements in turn affect the livelihood decisions of the households [24,25]. Consistently, the research on rural human settlements has been focused on urban building protection, settlement characteristics, spatial patterns, and the ecological environment [24,26–29]. Relatively little research has been carried out on the mechanism of how rural human settlements affect the employment and agricultural land transfer decisions of rural households from the perspective of household willingness and satisfaction.

In this study, we construct a theoretical framework that integrates rural human settlements and household livelihood capitals to explore the causal pathway of agricultural land transfer through rural employment. In addition, regional development imbalances always exist due to natural, economic, and social reasons. The levels of effectiveness and differences in unexpected agricultural land transfer decision-making between the eastern, central, and western regions have also attracted attention, and are affected by the location, nature, resources, and economic conditions. The study is driven by the following questions: (i) How do household livelihood capitals and rural human settlements affect the livelihood decisions related to employment and agricultural land transfer, respectively? (ii) What role do employment choices play in the process of agricultural land transfer? (iii) Is the causal pathway consistent in different regions?

## 2. Theoretical Framework

### 2.1. Agricultural Land Transfer Decision-Making Process

The SLF is adopted to develop our theoretical model [16]. The SLF emphasizes the role of livelihood capitals (human, natural, physical, financial, and social capitals) in livelihood decisions because they provide households with more opportunities to diversify livelihood strategies, thereby improving their capacity to cope with shocks and enhancing their livelihood sustainability [17]. In addition, the settlement conditions also affect the agricultural labor choices and agricultural land utilization [3,30]. Improvements in infrastructure, public services, and social welfare generally impose a stabilizing effect on the development of rural settlements [24,26]. In brief, households prefer to build their houses close to available infrastructure and services [31,32], which provides an incentive for agricultural land transfer. Specifically, employment choices may serve as an intermediary in the relationship between livelihood capitals and rural human settlements vs. agricultural land transfer. For example, when the rural living environment and living funds are not enough to support their production and livelihood, most households choose to seek out employment, meaning they have to transfer the agricultural land that they have no energy

to take care of. On the contrary, for farmers who choose to work and live in rural areas, the agricultural income will be a source of income that cannot be ignored.

Therefore, we develop a theoretical framework to reveal the mechanism of how household livelihood capitals and rural human settlements affect agricultural land transfer and regional differences, as shown in Figure 1.

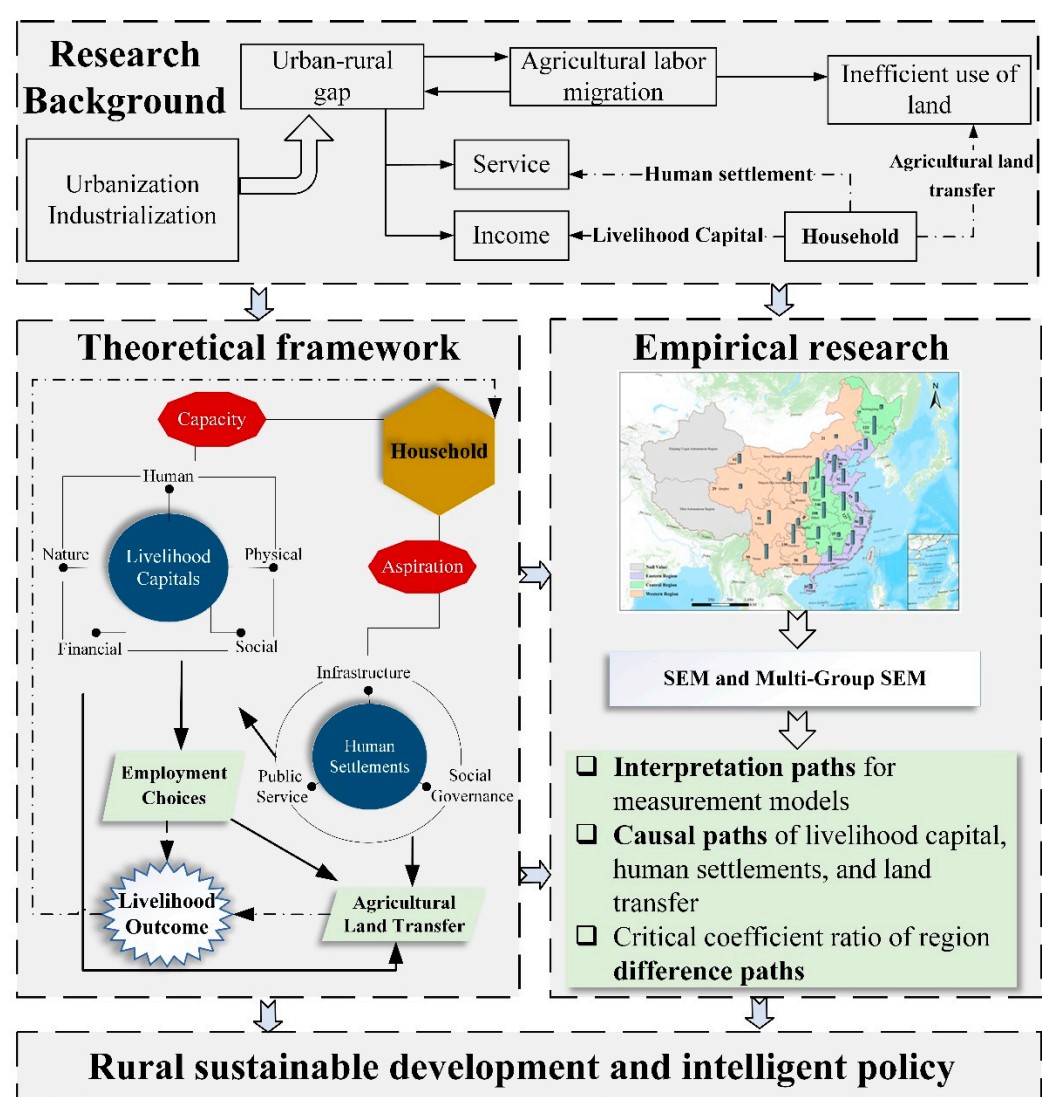

**Figure 1.** Agricultural land transfer decision-making process.

### 2.2. Dependent Variables: Agricultural Land Transfer

Faced with a dispersed and small-scale operation pattern, it is necessary to consider whether households are willing to adopt agricultural land transfer to combat the poor mechanical efficiency, wasted manpower, and increased costs caused by land fragmentation [33]. Moreover, households with no willingness or ability to farm will leave the agricultural land. In such cases, the agricultural land could be turned into financial capital through the outflow of agricultural land to achieve economic gains [34]. Thus, the agricultural land transfer is adopted as the dependent variable in this study. We use three variables to represent household agricultural land transfer: (i) which transfer behavior the household adopts; (ii) if adopted, what is the transfer area; (iii) if adopted, what is the transfer income.

### 2.3. Mediating Variable: Employment Choices

With the improvement in agricultural mechanization and labor productivity, the decline in the demand for agricultural labor has promoted the transfer of agricultural to non-agricultural industries [10]. Moreover, along with the acceleration of urbanization and the increase in urban employment opportunities, the non-agricultural transfer of agricultural labor continues to increase. When the economic conditions reach a certain level, the rural residents often choose to migrate to cities for better development opportunities [10]. Therefore, the employment choices of rural households can be divided into three categories, including agriculture, industry and commerce, and migratory work (Table 1).

**Table 1.** Mediating variables and dependent variables and indicators.

| Variables | Indicators | Description/Measurement |
|---|---|---|
| Employment Choices | Agricultural Work | Income from agricultural production and operation (see Appendix A, No.1) |
| | Industrial and Commercial | Income from industrial and commercial production (see Appendix A, No.2) |
| | Migratory Work | Migrant workers/household size |

### 2.4. Independent Variable: Household Livelihood Capitals

According to the SLF, an evaluation index system that measures the livelihood capital can be constructed, which includes five dimensions (human, natural, physical, financial, and social capitals) (Table 2).

**Table 2.** Explanatory variables and indicators.

| Variables | Indicators | Measurement |
|---|---|---|
| Infrastructure | Medical and Health Facilities | Satisfaction level for rural medical and health care (see Table A1, No.3) |
| | Service Facilities | Satisfaction level for rural services for the elderly, children, and the disabled (see Table A1, No.3) |
| Public Service | Employment Service | Satisfaction level for community labor employment services (see Table A1, No.3) |
| | Social Security Services | Satisfaction level for rural social security services (see Table A1, No.3) |
| Social Governance | Village Committee | Will government help be sought in case of dispute? (1 = Yes, 0 = No) |
| | Social Governance Satisfaction | Degree of help the village committee gives to the household (see Table A1, No.4) |
| Human Capital | Labor Availability | Labor force/household size |
| | Average Education | Total education years/household size |
| | Medical Treatment | The annual cost of health care |
| Natural capital | Agricultural Land Area | Total agricultural land area owned by household |
| | Cultivated Land Quality | Quality of cultivated land owned by household (see Table A1, No.5) |
| | Agricultural Land Use Type | Types of agricultural land owned by a household (see Table A1, No.6) |
| Physical Capital | Homestead Area | Area of homestead owned by household |
| | Durable Goods | Value of durable goods (see Table A1 No.7) |
| | Production Assets | Value of livestock and agricultural machinery in agricultural production and operation |

**Table 2.** *Cont.*

| Variables | Indicators | Measurement |
|---|---|---|
| Financial Capital | Government Subsidy | Amount of government subsidy (see Table A1, No.8) |
| | Household Debt | Amount of household debt (see Table A1, No.9) |
| | Financial Assets | Amount of household financial assets (see Table A1, No.10) |
| Social Capital | Village Cadre | Is there a family member serving as a village cadre? (1 = Yes, 0 = No) |
| | Cash Gift | Amount of gift (see Table A1, No.11) |
| | Social Security | Amount of social security (see Table A1, No.12) |

Specifically, the human capital refers to the labor ability, skills, and health status, which affect the livelihood strategies [16,35]. The variables representing human capital include labor availability, average education, and medical treatment. The development of the agricultural economy and the use of agricultural land results in the absorption and extrusion of the labor force [36]. The level of education and individual skillsets surfaced as important factors in most focus groups [37]. In addition, medical treatment constitutes an important dimension in human quality of life [17]. The natural capital represents the natural resources and services that households utilize [16], and is particularly important for households whose livelihoods rely on natural resources. Agricultural land is considered a determinant of livelihood decisions because it affects the potential income and food consumption of the household [38]. Therefore, this paper selected the agricultural land area, cultivated land quality, and agricultural land use type to represent the natural capital. The physical capital comprises the infrastructure and productive assets that facilitate household life and production [39]. It is composed of the homestead area, durable goods, and production assets in this paper. The homestead is the most important infrastructure for households [40]. Durable goods such as cars have a radical impact on the living style of the households [15,41]. Productive assets are investments made to improve the production efficiency [21,42]. The financial capital represents the financial resources that can be used to buy the goods necessary for survival and production [43]. Government subsidies, household debt, and financial assets are common sources of financial capital, which are selected in this paper. These resources provide support for livelihood activities and can be used to accumulate other livelihood assets [44]. The social capital refers to the resources that households can use to improve their livelihood capacity through social networks (such as kinship, friendship, neighbor relations), social organizations, or other groups (such as race or caste groups). It represents the social advocacy, social relations, subordination, and associations that households rely on when exploring various livelihood strategies [17]. As the most important social organization in China, the village committee helps households to enhance their livelihood [39]. The networks between relatives and friends are the primary channels for households to obtain information and assistance. Additionally, social security involves the redistribution of social resources, which is another important source of social capital [45]. Therefore, the village cadres, cash gifts, and social security are considered the evaluation variables of social capital.

*2.5. Independent Variable: Rural Human Settlements*

Rural human settlements can be categorized into material and non-material human settlements [24]. The infrastructure, public services, and social governance are important dimensions of rural human settlements [26,46]. The infrastructure refers to the material engineering facilities that provide convenience for the production and life of rural households [47]. Hospitals and clinics are the most basic and essential facilities needed to protect the health and life of the household [48]. Along with the elderly and children being increasingly left behind in rural areas, the infrastructure for specific groups directly affects the daily life experiences of the rural residents [49]. Therefore, medical and health facilities and

service facilities are taken as the component variables of rural infrastructure conditions. Public services guarantee household participation in social, economic, political, and cultural activities [50]. Promoting employment is an important way to improve household livelihood, which is the most relevant interest of households [41,51]. Social security services can provide material help for households who temporarily or permanently lose their working ability or face living difficulties [52]. Thus, employment services and social security services are used to measure public services in this study. Social governance is necessary to maintain social order, resolve social contradictions, and promote social equity [48].

As the executive organizations and management departments responsible for village affairs, the local government and village committees assist when households face livelihood difficulties. Their main tasks are to manage rural land and other properties, undertake the production services and coordination of the village, publicize national policies, and promote rural construction. They can further create a safe living environment by mediating disputes, maintaining social order, and managing public affairs [53]. In this paper, the village committee and social governance satisfaction are selected to represent social governance.

## 3. Materials and Methods

### 3.1. Household Survey and Data Source

The dataset used to analyze rural household agriculture land transfer is taken from the China Household Finance Studies (CHFS) in 2015. The dataset is representative in terms of both the economic development and geographic location. The random sampling survey employs computer-assisted personal interviews and a comprehensive quality assurance system to strictly control the measurement errors. Because agricultural land transfer mainly occurs in rural areas, this paper restricts the sample to rural households with agricultural land or agricultural land transfer behavior. Additionally, samples with extreme housing values are also excluded. The final sample includes 2089 households and involves 148 counties and 29 provinces.

In addition, in order to explore the regional differences in the causal pathways of land transfer decisions, this study divides the samples into three groups, the eastern, central, and western regions, according to natural, economic, social, and regional conditions. Among them, the eastern region has a flat terrain, rich aquatic and mineral resources, good agricultural production conditions, and strong economic vitality. Covering many plains, the central region is a major producer of grains and is rich in mineral resources and coal reserves, allowing the rapid development of heavy industry. The overall terrain in the western region is relatively high, and it involves plateau, desert, grassland, basin, and other landforms. Due to its long periods of cold weather, water shortages, and late development, its economic development and social governance levels are relatively lagging behind. According to the regional division, there are 653, 771, and 655 households in the eastern, central, and western regions, respectively (Figure 2).

### 3.2. The Structural Equation Model

This paper develops a structural equation model (SEM) to investigate the pathway of agricultural land transfer, which is widely used in behavioral sciences [24,54]. It estimates the relationships between multiple factors and derives the overall fitting degree, while the measurement error of the dependent and independent variables is permitted [55]. The SEM is composed of measurement models that measure the relationships between the observable variables and latent variables and structural models that measure the possible interactions among latent variables [56]. The observable variables are the directly measured variables that contain the raw data (agricultural land area, social security services, migratory work, etc.). In contrast, the latent variables cannot be directly measured but are manifested by observable variables (human capital, infrastructure, livelihood capital, etc.). The variable of employment choices is the mediating latent variable. The equations can be shown in the following forms:

$$x = \Lambda_x \xi + \delta, \tag{1}$$

$$y = \Lambda_y \eta + \varepsilon, \tag{2}$$

$$\eta = \beta \eta + \Gamma \xi + \zeta, \tag{3}$$

where x and ξ represent the exogenous observable variables and latent variables, respectively; y and η are the endogenous observable variables and latent variables, respectively; δ and ε are the independent measurement errors; $\Lambda_x$ and $\Lambda_y$ are the factor loads of indices x and y on ξ and η, respectively; β is the coefficient of the interaction between endogenous latent variables; Γ is the effective coefficient measuring the effect of exogenous latent variables on endogenous latent variables, and ζ is the residual.

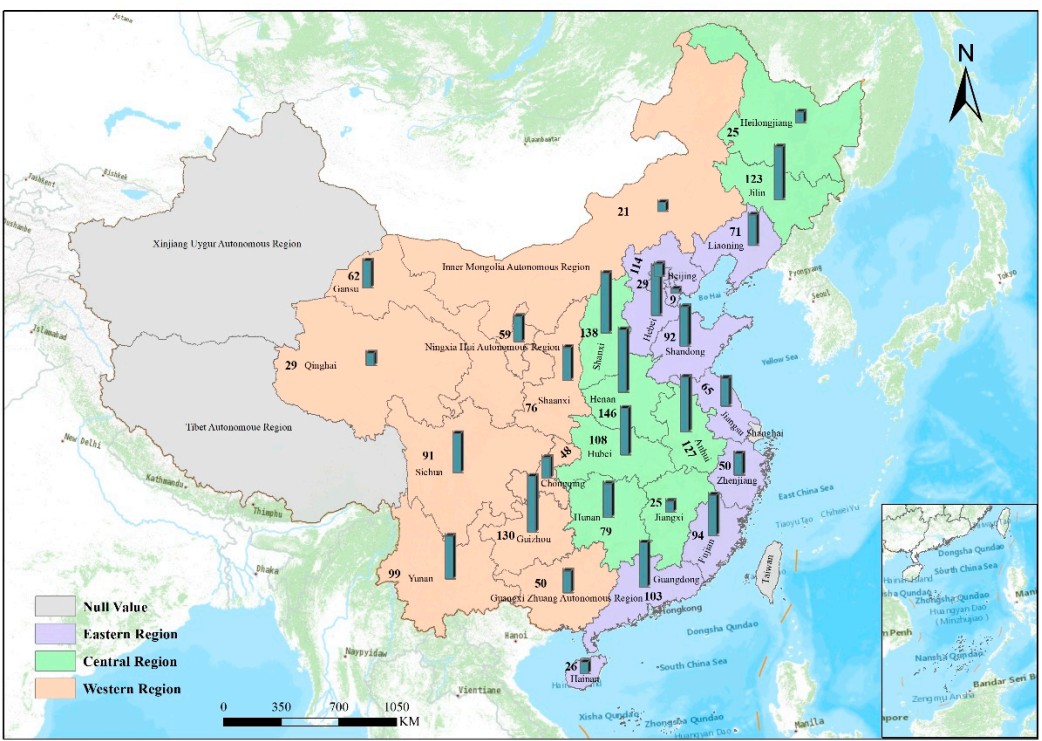

**Figure 2.** Distribution of household survey samples.

### 3.3. Multi-Group SEM

As an extension of the SEM, the multi-group SEM, is used to explore whether the research assumptions are consistent between different samples from a single scenario to a multi-group scenario [57]. The multi-group SEM includes three key steps: the inter-group invariance of the measurement model, the equivalencies of the structural model, and the analysis of difference paths [57]. The inter-group invariance of the measurement model and the equivalencies of the structural model are employed to confirm whether the causal path of the agricultural land transfer decision is statistically different at the regional scale [58]. The inter-group invariance of the measurement model is evaluated by limiting the measurement weight, structural covariances, and measurement residuals to be equal [58]. Similarly, the equivalencies of the structural model compare the differences in structural paths by constraining the measurement weight, structural weight, structural covariances, structural residuals, and measurement residuals [59]. The acceptable standard is *p* > 0.05 [59]. On this basis, the significance of the mediating effect and the path of the three regions were tested, and the results were in line with the standards [60].

## 4. Results

### 4.1. Descriptive Statistics

As shown in Figure 3, 15.31% and 12.54% of the households adopt the inflow and outflow decisions, respectively, while 4.79% of the households adopt both the inflow and

outflow decisions. The mean inflow area is 1.30 mu and the mean transfer expenditure is 0.39 thousand yuan.

**Figure 3.** Descriptive statistics of dependent variables.

Regarding the mediating variables and dependent variables, the sample households overall have a mean income of 6.47 thousand yuan from agricultural work, a mean income of 16.47 thousand yuan from industrial and commercial work, and a mean percentage of 26.02% migrants (Table 3).

**Table 3.** The descriptive statistics for the mediating variables.

| Indictor | Unit | Mean | Std. Dev | Min | Max |
|---|---|---|---|---|---|
| Agricultural Income | 1000 Yuan | 9.50 | 24.248 | 0.00 | 325.00 |
| Industrial and Commercial Income | 1000 Yuan | 2.65 | 24.19 | −500.00 | 300.00 |
| Migratory work | % | 27.95 | 20.60 | 0.00 | 100 |

Table 4, Figure 4a,b provide the descriptive statistics for the explanatory variables and regional differences in household livelihood and rural settlement. The data show that about 60% of the household members are contributing members, with the percentage being slightly higher in the central region. The average length of education received by household members is around 6 years, with the rate being relatively higher in the eastern region. The annual expenditure on medicines and health care for a household is 3.89 thousand yuan, which is more attributed to the western region. On average, the households only own one type of agricultural land (1.26 counts) and the quality of the cultivated land is low (average 2.90), although the type of agricultural land used in the western region and the quality of cultivated land in the eastern region have certain advantages. Traditional small-scale farming (average of 10.37 mu of agricultural land per household) increases the cost of land use and the difficulty of its management, especially in the eastern region. The household's access to physical capital shows large variations with the mean values for the homestead area (0.40 mu), durable goods (16.92 thousand yuan), and production assets (3.03 thousand yuan), being far from the maximum values. In the distribution

of regional physical capitals, the advantages of the central region are reflected in the homestead area, along with the superiority of durable goods in the eastern region and production assets in the western region. Similarly, the huge difference between the mean value and the maximum value for the financial capital variables, including government subsidies, household debt, and financial assets, shows that the general economic level of the rural households is relatively low, while there is a small number of households with high financial capital. Despite the relatively high level of average government subsidies in the central region, the financial advantages of the eastern region cannot be ignored. Regarding the social capital, 14.66% of the households have one or more family members working in the village committee. The average amount of cash gifts expended by each household is 3.12 thousand yuan, owing to the central region. The average social security is 4.26 thousand yuan, which provides more opportunities for a household to choose high-income livelihoods, especially in the eastern region. In terms of the rural human settlement conditions, the satisfaction levels for social security (3.48) and medical and health facilities (3.52) are the highest, followed by the satisfaction levels for service facilities (2.86) and social governance (2.84). In contrast, the satisfaction level for the rural employment services is the lowest (only 1.89). In terms of the regional comparison, the employment services and social security occupy a dominant position in the eastern region, accompanied by a relative lag in service facilities in the western region.

**Table 4.** Descriptive statistics for the explanatory variables.

| | Indictor | | Unit | Mean | Std. Dev | Min | Max |
|---|---|---|---|---|---|---|---|
| Livelihood Capitals | Human Capital | Labor Availability | % | 59.99 | 30.54 | 0.00 | 100.00 |
| | | Education | Years | 6.23 | 2.79 | 0.00 | 16.00 |
| | | Medical Treatment | 1000 Yuan | 6.24 | 2.89 | 0.00 | 12.47 |
| | Natural Capital | Agricultural Land Area | Mu | 9.55 | 15.97 | 0.00 | 204.00 |
| | | Cultivated Land Quality | Index | 2.89 | 1.44 | 0.00 | 5.00 |
| | | Agricultural Land Use Type | Counts | 1.18 | 0.64 | 0.00 | 5.00 |
| | Physical Capital | Homestead Area | Mu | 0.50 | 0.66 | 0.01 | 8.00 |
| | | Durable Goods | 1000 Yuan | 17.13 | 27.96 | 0.00 | 205.00 |
| | | Production Assets | 1000 Yuan | 3.09 | 8.10 | 0.00 | 80.10 |
| | Financial Capital | Government Subsidy | 1000 Yuan | 0.82 | 1.66 | 0.00 | 19.70 |
| | | Household Debt | 1000 Yuan | 4.09 | 18.45 | 0.00 | 240.00 |
| | | Financial Assets | 1000 Yuan | 17.11 | 38.25 | 0.00 | 364.05 |
| | Social Capital | Village Cadres | Index | 0.05 | 0.23 | 0.00 | 1.00 |
| | | Cash Gift | 1000 Yuan | 2.66 | 3.86 | 0.00 | 30.00 |
| | | Social Security | 1000 Yuan | 4.76 | 9.94 | 0.00 | 91.50 |
| Rural Human Settlements | Infrastructure conditions | Medical and Health Facilities | Index | 3.56 | 1.14 | 0.00 | 5.00 |
| | | Service Facilities | Index | 2.87 | 1.80 | 0.00 | 5.00 |
| | Public Service | Employment Service | Index | 1.42 | 1.88 | 0.00 | 5.00 |
| | | Social Security Services | Index | 3.73 | 1.03 | 0.00 | 5.00 |
| | Social Governance | Village Committee | Index | 0.06 | 0.23 | 0.00 | 1.00 |
| | | Social Governance Satisfaction | Index | 2.84 | 1.27 | 0.00 | 5.00 |

Mu is an area unit used in rural China; 1 mu = 1/15 ha.

### 4.2. Analysis of Measurement Models

The results of the SEM model show the causal relationship between the observed variables and the latent variables (Figure 5). The confirmatory factor analysis proves that the model is acceptable (Appendix B).

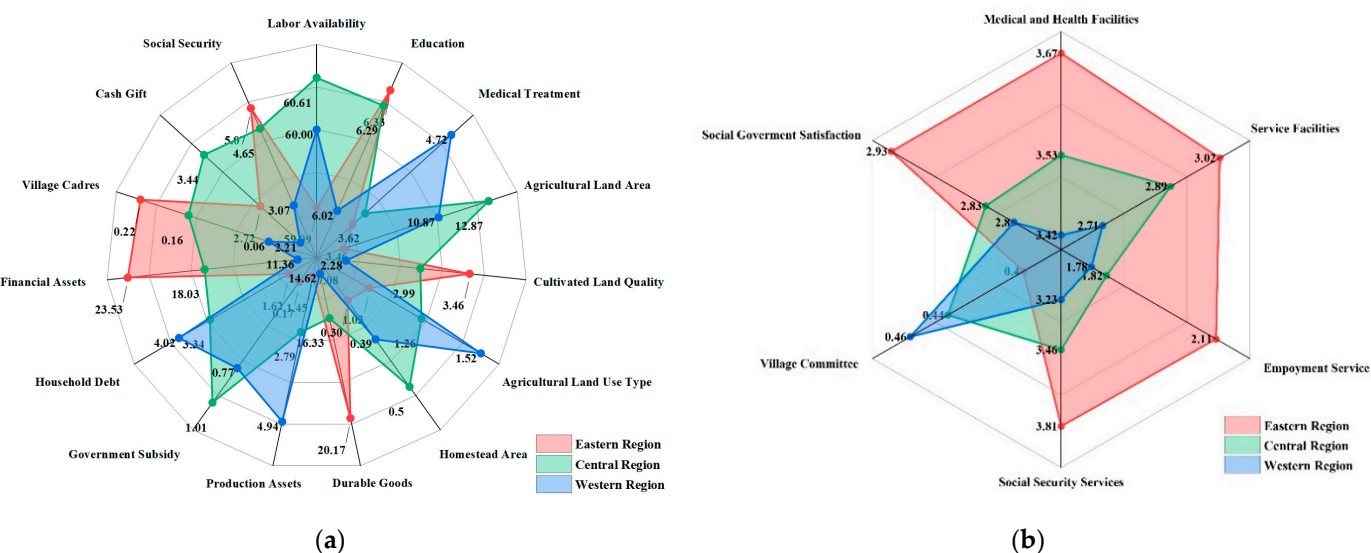

**Figure 4.** Descriptive statistics for the explanatory variables by region: (**a**) household livelihood; (**b**) rural settlement.

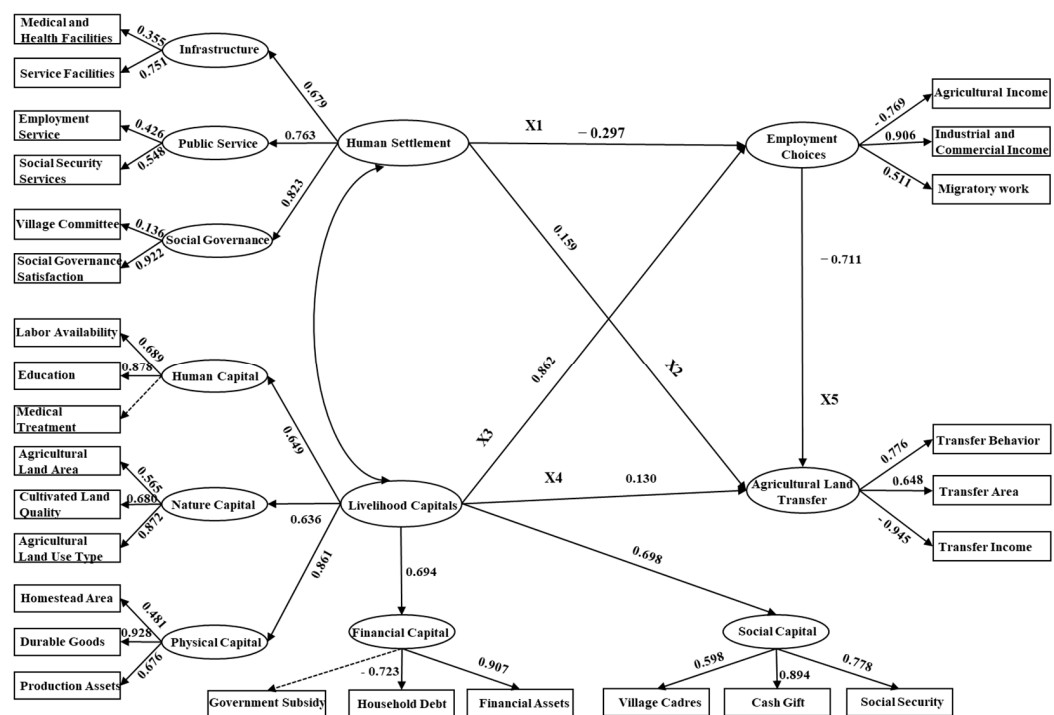

**Figure 5.** Parameter estimation results based on SEM.

4.2.1. Agricultural Land Transfer

The significant relationships show that the transfer behavior, transfer area, and transfer income can represent the agricultural land transfer well. Specifically, the transfer behavior has a significant positive effect on the agricultural land transfer, indicating that the behavior of the agricultural land inflow can improve the value of the agricultural land transfer. Since the transfer area equals the inflow area minus the outflow area, a high transfer area indicates a high inflow or a low outflow. As expected, there is a significant positive relationship between the transfer area and agricultural land transfer ($\beta = 0.648$, $p < 0.01$). Similarly, there is a significant negative correlation between the transfer income and agricultural land transfer ($\beta = -0.945$, $p < 0.01$). To sum up, the agricultural land transfer has a positive relationship with the agricultural land inflow.

### 4.2.2. Employment Choices

The results show that the three variables, namely the agriculture income, industrial and commercial income, and going out working, have statistically significant effects on the employment choices. The three variables have a consistent purpose, although they follow differently significant directions, which can be summarized as non-agricultural employment choices. Specifically, the relationship between the agricultural income and employment choices is negative and significant ($\beta = -0.769$, $p < 0.01$), while the relationship between the industrial and commercial income and migratory work is positive ($\beta = 0.906$, $p < 0.01$). Consequently, the values of the latent variables of rural employment choices will be greater if a household has lower agricultural income, higher industrial and commercial income, and a larger proportion of migrants.

### 4.2.3. Livelihood Capitals

There is a significant positive correlation between human capital and livelihood capital, which indicates that the households' livelihood capital increases with their human capital when other variables are controlled. For the human capital, the labor availability is a key variable ($\beta = 0.689$, $p < 0.01$), which determines the consumption and production of the household. The education level of the household members has significant associations with the human capital ($\beta = 0.878$, $p < 0.01$). Here, the relationship between the health status and human capital is not significant.

A statistically significant relationship can be found between the natural capital and livelihood capital ($\beta = 0.636$, $p < 0.01$). Households that possess larger quantities, better quality, and more types of agricultural land have higher stocks of natural capital. High quality usually means that more crop products can be produced. Additionally, the use of multiple crop cultivation approaches creates more opportunities to improve the livelihood capital, as market access largely determines crop selection.

The physical capital also has a significant positive relationship with household livelihood capitals ($\beta = 0.861$, $p < 0.01$). The homestead can play a positive role when the household livelihood capitals experience negative shocks. Durable goods allow households to find production channels or livelihood opportunities and make it easier to transport agricultural products. The ownership of production assets enhances the grain production capacity, thereby contributing to the management of larger or more agricultural land.

The financial capital is indispensable for the livelihood strategy of households, as proven by the statistically significant relationship between the financial capital and livelihood capital ($\beta = 0.694$, $p < 0.01$). The government subsidy is not significantly associated with the financial capital, perhaps because it merely meets the basic survival needs. Household debt has negative effects on the financial capital. Households with debts, especially poorer households, can severely impede their capacity to make economic and social choices regarding their precarious livelihoods.

The social capital has a positive relationship with the household livelihood capital ($\beta = 0.698$, $p < 0.01$). The significant relationship between the village cadres and social capital may reflect the fact that the cadres have a certain advantage over ordinary migrant rural workers when acquiring information ($\beta = 0.598$, $p < 0.01$). Households with greater social connectedness are more likely to learn about the opportunities for livelihood capital accumulation. Additionally, social security can enhance the social capital ($\beta = 0.778$, $p < 0.01$), thereby situating livelihood capitals under the broader umbrella of risk management.

### 4.2.4. Rural Human Settlements

The trend towards rural human settlements can be explained by their increased infrastructure, public services, and social governance, all of which show significant effects. Improvements in infrastructure can promote settlement conditions, as reflected by the positive values ($\beta = 0.679$, $p < 0.01$), explaining the changes in location of settlements from resource-based to facilities-based areas. Medical and health facilities are important driving factors to improve the rural human settlement conditions ($\beta = 0.355$, $p < 0.01$). Service

facilities for the care of the elderly, children, and the disabled can support the sustainable development of rural human settlements for the protection of rural elderly individuals and children who have been left-behind. Their quality of life increasingly depends on public services, especially high-quality employment and social security services ($\beta = 0.763$, $p < 0.01$). Employment information and services are very important for rural residents, which could reduce the loss of the rural population and promote employment. The promotion of social security services is also an effective way to protect low-income households and alleviate social contradictions, which is conducive to improving the rural human settlement conditions. As an effective way to reflect the quality of social services, social governance is an important way of improving the settlement conditions ($\beta = 0.823$, $p < 0.01$). If a household can take the initiative to seek help from state bodies and achieve satisfactory solutions to production and life difficulty issues or conflicts, it can effectively improve its living stability to create safe and comfortable settlement conditions.

### 4.3. Causal Pathway of Agricultural Land Transfer Decisions

This paper hypothesizes that the influence of livelihood capitals and human settlements on agricultural land transfer can be divided into two pathways—the direct impact path and the indirect impact path, as mediated by employment choices. The SEM provides reliable evidence for the theoretical hypothesis (Table 5).

**Table 5.** Direct, indirect, and total effects of livelihood capitals and human settlements on agricultural land transfer.

| Path | β | SE |
|---|---|---|
| Human Settlements->Agricultural Land Transfer | 0.159 | 0.029 |
| Human Settlements->Employment Choices->Agricultural Land Transfer | 0.211 | 0.022 |
| Human Settlements and Employment Choices->Agricultural Land Transfer | 0.370 | 0.030 |
| Livelihood capitals->Agricultural Land Transfer | 0.130 | 0.051 |
| Livelihood Capitals-> Employment Choices->Agricultural Land Transfer | −0.613 | 0.057 |
| Livelihood Capitals and Employment Choices->Agricultural Land Transfer | −0.483 | 0.038 |
| Human Settlements, Livelihood Capitals, and Employment Choices->Agricultural Land Transfer | −0.113 | 0.039 |

#### 4.3.1. Impacts of Livelihood Capitals on Agricultural Land Transfer

The direct relationship between the livelihood capital and agricultural land transfer is primarily positive and significant ($\beta = 0.130$, $p < 0.01$). It shows that households with high livelihood capitals have a certain preference for land inflow. In terms of the indirect path, the livelihood of rural households has a significant negative impact on the path of agricultural land transfer through the mediating factor of rural employment choices ($\beta = -0.613$, $p < 0.01$), which indicates that the livelihood capitals tend to promote the outflow of the population from agriculture to obtain higher capital. Superior livelihood capitals not only support the inflow of agricultural land but also provide more opportunities for households to turn to industry and commerce or migration. The moderator–mediator test synthesizes the results of the direct and indirect paths, showing a significant negative impact on the whole ($\beta = -0.483$, $p < 0.01$). Compared with obtaining more agricultural income, households with superior livelihood resources are more likely to choose non-agricultural employment, which can lead to more funds and a better quality of life, thereby causing an overall trend of agricultural land outflow.

#### 4.3.2. Impact of Rural Human Settlements on Agricultural Land Transfer

The direct effect of the rural human settlement conditions on agricultural land transfer is positive and significant ($\beta = 0.159$, $p < 0.01$). It also shows that a superior living environment is conducive to promoting the inflow of agricultural land to a certain extent.

As expected, the rural human settlement conditions exert a significant positive impact on agricultural land transfer through employment choices ($\beta$ = 0.211, $p$ < 0.01), indicating that a comfortable and convenient living environment can retain rural residents, thereby investing the labor force into agricultural land resources. The combined effect of the direct and indirect paths intensifies the positive impact of the rural human settlement conditions on agricultural land inflow ($\beta$ = 0.370, $p$ < 0.01), which proves the significant role of rural human settlement improvements in agricultural land use.

### 4.3.3. Impacts of Livelihood Capitals and Rural Human Settlements on Agricultural Land Transfer

The synthesis influence of livelihood capitals and rural human settlement conditions on agricultural land transfer is negative and significant ($\beta$ = −0.113, $p$ < 0.01). This result indicates that improving the living standards and settlement conditions is more conducive to agricultural production. The path to promote agricultural land inflow includes the direct impact of the livelihood capital, the indirect impact of the human settlements, and the direct and comprehensive impact of the human settlements. However, the indirect impact and comprehensive impact path of the livelihood capitals can promote the outflow of agricultural land. Hence, the results can provide insights for improving the agricultural land use efficiency and activating rural vitality.

### 4.4. Regional Differences in Path Coefficients

The multi-group SEM is used here to explore the consistency of the theoretical model among the different regional groups; that is, to test whether the causal path coefficients of the settlement conditions, livelihood capitals, and agricultural land transfer in the household clusters living in the eastern, central, and western regions are equal, as well as the mediating role of employment choices. It is necessary to evaluate the inter-group invariance of the measurement model and the equivalencies of the structural model. As shown in Appendix C and Table A5, there is no significant difference between groups in the measurement model, which means that the interpretation of the observed variable to potential variables spans the regional characteristics. It is worth noting that the path coefficients of the structural model (Figure 6) show significant differences among regions (Table A6).

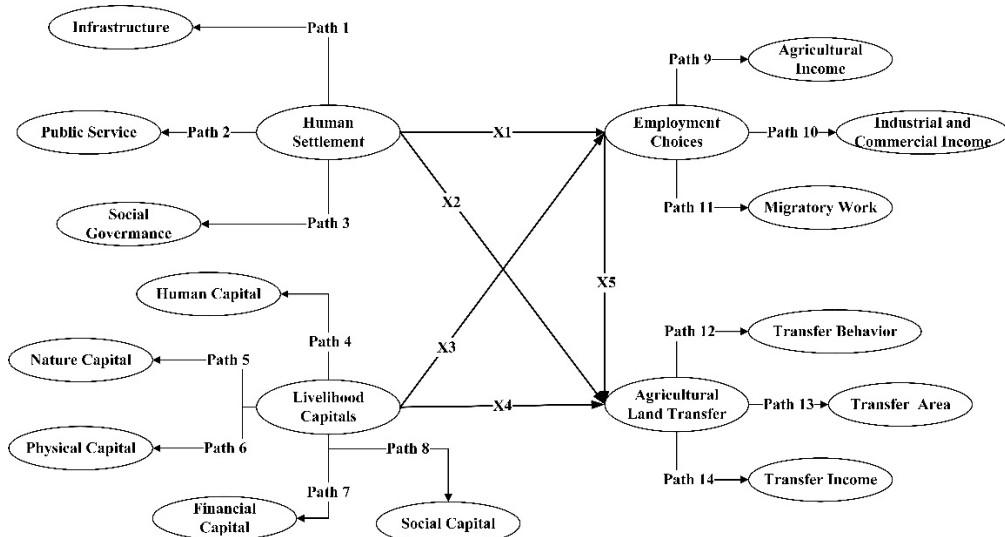

**Figure 6.** Structural models and paths.

To further locate the path, the critical ratio is employed. Its significance is used to prove the differences among regions of the same path, including the eastern region vs. central region, eastern region vs. western region, and central region vs. western region. As shown in Tables 6 and 7, the social governance in the western region plays a relatively weak role. The critical ratio and path coefficient of the natural capital in the three regions jointly express its prominent position in the central region, followed by the western region and the eastern region. The possible drivers are the abundant arable land resources in the central region and the dependence of the livelihood patterns on the natural resources in the western region. Relatively speaking, households living in the central region have obvious advantages in terms of their physical capital, possibly because they have more convenient conditions for mechanized farming and transportation. The advantages in terms of financial and social capital for households living in the eastern region is that they may profit from this economic development background, which has more employment opportunities and channels to promote capital accumulation. There are inverse differences in the roles of a migrant worker in the western region, indicating that the household tends to work locally even if it is far away from agriculture. Notably, the improvement in settlement conditions in the western region will promote such an exodus, and the accumulation of livelihood capitals will increase the incentives to stay away from agriculture. For households living in the eastern region, improved settlement conditions can increase the attractiveness of agricultural land.

**Table 6.** The structural model path coefficients for different regions.

| No. | Path | Eastern Region | Central Region | Western Region |
|---|---|---|---|---|
| 1 | Settlements Conditions->Infrastructure | 0.454 | 0.457 | 0.523 |
| 2 | Settlements Conditions->Public Service | 0.648 | 0.669 | 0.735 |
| 3 | Settlements Conditions->Social Governance | 0.728 | 0.768 | 0.661 |
| 4 | Livelihood Capitals->Human Capital | 0.420 | 0.464 | 0.461 |
| 5 | Livelihood Capitals->Natural Capital | 0.124 | 0.337 | 0.272 |
| 6 | Livelihood Capitals->Physical Capital | 0.724 | 0.802 | 0.771 |
| 7 | Livelihood Capitals->Financial Capital | 0.444 | 0.562 | 0.508 |
| 8 | Livelihood Capitals->Social Capital | 0.476 | 0.516 | 0.513 |
| 9 | Employment Choices->Agricultural Income | −0.353 | −0.364 | −0.407 |
| 10 | Employment Choices->Industrial and Commercial Income | 0.476 | 0.487 | 0.512 |
| 11 | Employment Choices-> Migratory Work | −0.033 | −0.055 | 0.047 |
| 12 | Agricultural Land Transfer->Transfer Behavior | 0.832 | 0.826 | 0.813 |
| 13 | Agricultural Land Transfer->Transfer Area | 0.657 | 0.621 | 0.647 |
| 14 | Agricultural Land Transfer->Transfer Income | −0.972 | −0.957 | −0.947 |
| X1 | Settlements Conditions->Employment Choices | −0.304 | −0.302 | 0.240 |
| X2 | Settlements Conditions->Agricultural Land Transfer | 0.227 | 0.203 | 0.139 |
| X3 | Livelihood Capitals->Employment Choices | 0.712 | 0.822 | 0.848 |
| X4 | Livelihood Capitals->Agricultural Land Transfer | 0.051 | 0.122 | 0.153 |
| X5 | Employment Choices->Agricultural Land Transfer | −0.795 | −0.834 | −0.853 |

From a comprehensive perspective, the settlement conditions can significantly promote agricultural land inflow in the eastern and central regions, either directly or indirectly through employment choices (Table A7). On the contrary, the livelihood capital can promote agricultural land outflow. In addition, the role of the settlement conditions and livelihood capitals in the eastern region in agricultural land transfer extends beyond the central region. When overlaying the livelihood capitals, the willingness to flow out of the agricultural land in the western region is even more obvious because the improved settlement conditions cannot keep households from devoting themselves to agriculture (Table 8).

**Table 7.** The critical path coefficient ratios of regional differences.

| Path | Eastern vs. Central Region | Eastern vs. Western Region | Central vs. Western Region |
|------|---------------------------|----------------------------|---------------------------|
| 1 | 0.748 | 0.753 | 0.022 |
| 2 | 0.293 | 0.823 | 0.229 |
| 3 | 0.757 | −2.324 *** | −2.275 *** |
| 4 | −1.336 | −1.137 | 0.601 |
| 5 | 2.101 *** | 1.972 *** | 2.079 *** |
| 6 | −1.987 *** | −1.183 | 3.655 *** |
| 7 | −2.149 *** | −1.964 *** | 0.492 |
| 8 | −1.962 *** | −2.562 *** | 0.891 |
| 9 | −0.032 | −0.413 | −0.406 |
| 10 | 0.762 | 0.637 | 1.402 |
| 11 | −0.351 | 2.272 *** | 2.691 *** |
| 12 | 0.514 | 1.775 | 1.427 |
| 13 | 0.486 | −1.898 | −1.594 |
| 14 | −0.079 | 1.068 | 1.292 |
| X1 | −0.672 | 5.189 *** | 4.687 *** |
| X2 | −3.604 *** | −3.718 *** | 0.324 |
| X3 | −0.190 | 0.863 | 1.209 |
| X4 | 1.356 | −2.094 *** | −2.610 *** |
| X5 | 0.629 | −0.794 | −5.181 |

*** is $p < 0.001$.

**Table 8.** The direct, indirect, and total effects of livelihood capitals and rural human settlements on agricultural land transfer in different regions.

| Path | Eastern Region | Central Region | Western Region |
|------|----------------|----------------|----------------|
| Settlements Conditions->Agricultural Land Transfer | 0.227 | 0.203 | 0.139 |
| Settlements Conditions->Employment Choices->Agricultural Land Transfer | 0.242 | 0.252 | −0.205 |
| Settlements Conditions and Employment Choices->Agricultural Land Transfer | 0.469 | 0.455 | −0.066 |
| Livelihood Capitals->Agricultural Land Transfer | 0.051 | 0.122 | 0.153 |
| Livelihood Capitals and Employment Choices->Agricultural Land Transfer | −0.566 | −0.686 | −0.723 |
| Livelihood Capitals->Employment Choices->Agricultural Land Transfer | −0.515 | −0.564 | −0.570 |
| Settlements Conditions, Livelihood Capitals, and Employment Choices->Agricultural Land Transfer | −0.046 | −0.109 | −0.636 |

## 5. Discussion

It is widely believed that agricultural land transfer or abandonment is an irrepressible socio-economic phenomenon, as non-agricultural income is more profitable than agricultural income [3,61]. However, earth-shaking changes have taken place in rural settlements with the improvement of infrastructure, acceleration of urban and rural transportation networks, and popularization of the Internet [26]. This paper assumes that the human settlement conditions and livelihood capitals can influence the decision-making regarding agricultural land transfer. Moreover, employment choice plays a mediating role in this process. Therefore, taking the human settlement conditions and livelihood capitals as independent variables, employment choice as mediating variable, and agricultural land transfer as the dependent variable, this paper uses the SEM to explore the causal pathways and their regional differences. It provides important insights for policy-makers to increase the value of agricultural land and enhance the rural vitality.

### 5.1. The Mediating Role of Employment Choices in the Process of the Agricultural Land Transfer

Rural residents will allocate their labor based on the comparative income provided by agricultural and non-agricultural industries. When this situation occurs, households choose to spend less time on farming, resulting in agricultural land transfer. With China's rapid economic development, it is controversial whether the increase in non-agricultural

income will reduce or promote households' investment in agriculture. Most studies believe that the non-agricultural conversion of rural labor is the most fundamental driving force of agricultural land transfer [13,33]. However, some studies have pointed out that the agricultural land transfer lags far behind the non-agricultural transfer of the rural labor force [12]. The reason why many non-agricultural laborers did not give up agricultural management rights may be that migrant workers with rural hukou cannot enjoy the same rights and social security benefits as urban hukou under the urban–rural binary household registration system in China. In addition, migrant workers tend to retain their agricultural land management rights as a basic safety net due to their unstable jobs. There is no doubt that the allocation of the rural labor force will change the allocation of agricultural rural land resources. According to this paper, an increase in livelihood capital will enable households to access more employment opportunities, thereby promoting non-agricultural employment and leading to the outflow of agricultural land. When households living in rural areas have the same social welfare systems and living environments as the cities, they are more willing to stay in rural areas to accompany their family members. In this case, the households' dependence on agricultural land will increase. Therefore, the improvement of rural human settlements will promote the inflow of agricultural land by restraining the outflow of rural labor.

### 5.2. Interaction Effect of Rural Human Settlements and Livelihood Capitals on Agricultural Land Transfer

A rural settlement is a multi-dimensional integrated system formed by the interactions of various elements. Land elements play a fundamental role in providing resource support for rural revitalization and space for the development of rural industrial development. Agricultural land use is an important issue related to household livelihoods and agricultural development [62]. According to the agricultural household economic models, rural households, as the decision-makers for their agricultural land, take the maximization of family utility as the goal in the decision-making related to agricultural land production and management. The maximization of family utility is determined by the optimization of their production and quality of life [63]. Capital accumulation is the main goal in production optimization, and it is also the fundamental reason for agricultural land transfer. The integration of "resources, capital, and assets" would be a crucial way to promote rural development, improve the living environment, and provide a good business environment for agricultural industrialization. Some studies believe that households have more economic freedom and that their willingness to transfer agricultural land also increases with the increase in capital accumulation [12]. However, this study finds that the capital accumulation will promote the inflow of agricultural land because the capital can also be used as the cost of the land inflow in developing industries. Based on the processes of urbanization and economic development, the internal and external environmental conditions for rural and agricultural development are undergoing major changes in China. As an effective measure of change in agricultural production and household living environments, the rural human settlement transformation is highly valued by managers. It can be seen from the research results that the improvement in rural human settlement conditions will not only attract households to stay in rural areas and promote the inflow of agricultural land but will also directly strengthen the inflow of agricultural land, so as to allow the reallocation of agricultural land resources.

### 5.3. Regional Differences and Policy Recommendations

Based on the relationships between human settlements, livelihood capitals, employment choices, and agricultural land transfer explored in this study, the guidance, support, and restrictive policies offered by local governments play an important role in agricultural land use and rural revitalization. Facing the magnification of human settlement conditions and livelihood capitals in the central region and the unexpected role of the human settlement conditions in promoting the outflow of agricultural land in the western region, the

formulation of policies such as the improvement of human settlements and land use should be carefully considered. In general, improving the efficiency of land use and protecting the agricultural land are the designated goals of such policies. More importantly, the actual needs of the farmers, who are the main actors in rural areas, and the direct beneficiaries should be considered in the policy design.

### 5.3.1. Attracting Talent to Return to Rural Areas

As the most important factor in rural development, rural–urban migration results in a series of socio-economic changes, including changes to the labor market, rural restructuring, and balanced regional development. The loss of the rural population, generally the most active, youngest, and highest quality sector of the labor force, has not only affected the age structure but has also changed the intellectual structure. The shortage of talent, less-educated human resources, and weakened main body of development caused by it have negative impacts on the development of modern agriculture and the popularization of agricultural science and technology. To alleviate these dilemmas, policy instruments that encourage and guide talent to return to the rural areas should be implemented. It is essential to specify supporting policies to attract and retain talent for long-term rural development. First, it is necessary to provide enterprise education and training on modern science and technology, modes of production, business philosophy, and practical technology for those return migrants, so as to cultivate them into new professional farmers to meet the development needs. Second, financial support is absolutely necessary. The government can give financial subsidies for innovation and entrepreneurship projects for return migrants, provide loans for agricultural moderate-scale operation, and implement a policy of tax and fee reductions.

### 5.3.2. Improving the Rural Human Settlements

With the development of the rural economy and the improvement of rural livelihoods, rural households are gradually giving up their traditional way of life and seeking a diversified lifestyle that covers employment, communication, leisure, entertainment, and tourism. In order to make better use of agricultural land and maintain the rural vitality, it is necessary to improve the rural human resettlement conditions in rural areas. Such conditions are an important source of rural residents' happiness and sense of achievement, which is a guarantee for better living standards and better quality of life for rural residents. Infrastructure, public services, and social governance are the main aspects of rural human settlements. First, the government should promote the extension of urban infrastructure construction to rural areas, including transportation, water conservancy, and energy projects, so as to promote the upgrading of rural infrastructure. To ensure the development of rural areas, the construction of digital rural facilities is urgent. More importantly, the gap between rural and urban areas in public education, health care, pension services, and social security should be gradually eliminated to ensure the equalization of basic public services between urban and rural areas. The promotion of cultural services based on folk art and group activities is an important way to enrich the spiritual and cultural lives of rural residents. In addition, there is still more room for the government to improve the social governance in rural areas, such as by increasing the ability of the grassroots cadres, emphasizing legal literacy, and improving the quality of life for rural residents.

### 5.3.3. Enhancing Household Livelihood Capitals

Ensuring the basic and long-term livelihoods has become the central issue in balancing urban–rural development and rural revitalization in China. Livelihood capitals play important roles in rural restructuring and household income growth, affecting the human non-agriculturalization, industrial cultivation, and land use transition processes and the rural self-development ability. The livelihood capitals of households, therefore, should be fundamentally promoted. The government should promote the transformation of the agricultural industrial structure, enhance the strength of the regional economic

development, and improve the sustainability of the livelihood capital growth. Exploring the use of agricultural and rural resources and developing characteristic industries using new ideas, technologies, and channels are the predominant ways to improve the livelihood capitals, which should be encouraged and supported by the government. Land is the spatial carrier of rural industrial development. However, the rural areas are experiencing depopulation and housing modernization, which have led to the abandonment of agricultural land resources. Thus, the government should provide development spaces for modern agriculture and other rural industries, which would improve the household livelihood capital through consolidating the inefficiently utilized land, promoting land use circulation, and appropriate scale management.

## 6. Conclusions

From a macro-perspective, agricultural land transfer is an effective way to alleviate land abandonment and improve land use efficiency, and can contribute to food security and social stability. From a micro-perspective, agricultural land transfer can reduce the cost of agricultural production and change household livelihoods. Based on the dataset from the China Household Finance Studies (CHFS), this paper explored the impacts of livelihood capitals and rural human settlements on agricultural land transfer and regional differences using a structural equation model. The mediating role of employment choices was also examined.

The livelihood capitals were further divided into human, natural, physical, financial, and social capitals according to the sustainable livelihood framework. The results showed that the accumulation of livelihood capital can directly increase the possibility of agricultural land inflow, and can also indirectly reduce the dependence on agricultural land by stimulating non-agricultural employment. The rural human settlements were measured from three dimensions, including infrastructure, public services, and social security. The results indicated that the improvement of rural human settlement conditions can promote the inflow of agricultural land, but can also indirectly strengthen the willingness of households to flow into agricultural land by creating a comfortable living environment and restraining the outflow of the population. In the context of rural revitalization, the improvement of rural human settlements and household livelihood capitals can effectively promote agricultural land transfer, which could accelerate the transfer of agricultural land from non-agricultural households to those with farming willingness, except for in the western region. Therefore, it is necessary to fully grasp the internal mechanism involved when making policies. There is also a practical need to attract talent to return to the rural areas and provide them with training and financial support to develop modern agriculture, so as to promote the transformation of agricultural land use from traditional extensive agriculture to intensive modern agriculture.

**Author Contributions:** Conceptualization, W.W., J.G. and Y.W.; methodology, W.W.; software, W.W. and Y.S.; validation, W.W.; formal analysis, W.W.; investigation, W.W. and Y.W.; resources, Y.W.; data curation, W.W.; writing—original draft preparation, W.W.; writing—review and editing, W.W., J.G. and Y.W.; visualization, W.W.; supervision, J.G. and Y.W.; project administration, J.G. and Y.W.; funding acquisition, J.G. All authors have read and agreed to the published version of the manuscript.

**Funding:** This research was funded by the National Natural Science Foundation of China (Grant No. 41901213, 42071254), the Natural Science Foundation of Hubei Province (Grant No. 2020CFB856), and the Fundamental Research Funds for the Central Universities, China University of Geosciences (Wuhan) (Grant No. 26420190065, 26420180052).

**Data Availability Statement:** Not applicable.

**Acknowledgments:** We are very grateful for the household survey conducted by the Southwest University of Finance and Economics. Finally, we are grateful to the household interviewed for their time and data.

**Conflicts of Interest:** The authors declare no conflict of interest.

## Appendix A

**Table A1.** Scope of the explanatory variable.

| No | Indictor | Scope |
|----|----------|-------|
| 1 | Agricultural production and operation | Cultivate food crops; Cultivate economic crops; Plant and transport trees; Raise livestock and poultry; Breed and fish aquaculture; Cultivate other crops |
| 2 | Industrial and commercial production | Self-employed/industrial and commercial enterprises; Joint-stock limited company; Limited liability company; Partnership; Sole proprietorship; No formal form of organization; Others |
| 3 | Satisfaction | 1 = very dissatisfied; 2 = not very satisfied; 3 = average; 4 = fairly satisfied; 5 = very satisfied |
| 4 | Help degree | 1 = none; 2 = not too large; 3 = average; 4 = relatively large; 5 = very large |
| 5 | Cultivated land quality | 1 = very poor; 2 = poor; 3 = average; 4 = good; 5 = very good |
| 6 | Agricultural land use type | Cultivated land; woodland; grassland; garden; others |
| 7 | Durable Goods | Car; camera/camera; TV; Washing machine; Refrigerator; Air conditioner; Computer; Audio; Water heater; Furniture; Musical instrument; Mobile phone; Induction cooker; Microwave oven; Water dispenser; Others |
| 8 | Government subsidy | Special poverty allowance; Only child award; Five guarantees allowance; Pension; Relief/disaster relief fund; Food subsidy; Grain for green; Subsistence allowance; Education subsidy; Housing subsidy; Agricultural subsidy; Others |
| 9 | Household Debt | Education debt; Medical debt; Credit card; Others |
| 10 | Financial Assets | Current deposits; Fixed deposits; Stocks; Funds; Financial products; Bonds; Derivatives; Non-RMB assets; Precious metals; Cash; Others |
| 11 | Cash Gift | Holiday expenses; Red and white happy expenses; Others |
| 12 | Social Security | Social endowment insurance and enterprise annuity; Medical insurance; Unemployment insurance; Housing accumulation fund; Industrial and commercial insurance; Maternity insurance; Others |

## Appendix B

A confirmatory factor analysis (CFA) is employed to confirm whether the data compiled here have a good fit for the model estimation by measuring the convergent validity, discriminant validity, and construct validity. Specifically, the convergent validity evaluates whether the observable variables can fully explain each latent variable and analyze the internal consistency of the variables. The average variance extracted (AVE, greater than 0.5) and the composite reliability (CR, greater than 0.7) are commonly used to determine the degree of convergent validity [64]. The discriminant validity is measured via the factor correlations of the latent variable, which reflect the independence of the different latent variables. The discriminant validity is satisfactory if its value exceeds the square root of the AVE (Table A2). The construct validity of the model can be evaluated via model fit indices (Table A3) [65]. In addition, to test the significance of the impacts of rural human settlements and livelihood capitals on agricultural land transfer as mediated by employment choices, in this paper we conduct a moderator–mediator test with a 95% confidence interval (Table A4) [66,67]. The path is significant when there is no zero between the lower and upper limit of the bias-corrected 95% CI and percentile 95% CI [67].

**Table A2.** Confirmatory factor analysis (CFA) of the discriminant validity.

| | AVE | CR | Human Capital | Natural Capital | Physical Capital | Financial Capital | Social Capital | Infrastructure | Public Service | Social Governance |
|---|---|---|---|---|---|---|---|---|---|---|
| Human capital | 0.501 | 0.741 | 0.708 | | | | | | | |
| Natural capital | 0.514 | 0.755 | 0.011 *** | 0.717 | | | | | | |
| Physical capital | 0.517 | 0.750 | 0.127 *** | 0.057 *** | 0.719 | | | | | |
| Financial capital | 0.533 | 0.764 | −0.027 *** | −0.007 *** | −0.109 *** | 0.730 | | | | |
| Social capital | 0.587 | 0.806 | 0.044 *** | 0.028 *** | 0.147 *** | −0.028 *** | 0.766 | | | |
| Infrastructure | 0.621 | 0.761 | 0.034 *** | 0.018 *** | 0.058 *** | 0.038 *** | 0.038 *** | 0.788 | | |
| Public service | 0.592 | 0.742 | 0.048 *** | 0.025 *** | 0.081 *** | 0.052 *** | 0.053 *** | 0.280 *** | 0.769 | |
| Social governance | 0.577 | 0.722 | 0.052 *** | 0.027 *** | 0.088 *** | 0.057 *** | 0.058 *** | 0.336 *** | 0.396 *** | 0.760 |
| The square root of AVE | | | 0.708 | 0.717 | 0.719 | 0.730 | 0.766 | 0.788 | 0.769 | 0.760 |

The square root of the AVE is shown in bold on diagonals. Off the diagonals are Pearson correlations of constructs. The discriminant validity is achieved when the diagonal value in bold is higher than the values in its row and column. *** is $p < 0.001$.

**Table A3.** Confirmatory factor analysis (CFA) of the discriminant validity.

| GOF Measures | $\chi^2/df$ | CFI | GFI | AGFI | NFI | IFI | RMSEA |
|---|---|---|---|---|---|---|---|
| Recommended levels | <5.000 | >0.900 | >0.900 | >0.900 | >0.900 | >0.900 | <0.050 |
| Test value | 4.894 | 0.945 | 0.976 | 0.965 | 0.932 | 0.945 | 0.044 |
| Result | Pass | Pass | Pass | Pass | Pass | Pass | Pass |

CFI is the comparative fit index; GFI is the goodness-of-fit index; AGFI is the adjusted goodness-of-fit index; NFI is the normed fit index; IFI is the incremental fit index; RMSEA is the root mean square error of approximation.

**Table A4.** Confirmatory factor analysis (CFA) of the discriminant validity.

| Path | Bias-Corrected 95% CI | | Percentile 95% CI | |
|---|---|---|---|---|
| | Lower | Upper | Lower | Upper |
| Settlements Conditions->Agricultural Land Transfer | 0.102 | 0.213 | 0.105 | 0.217 |
| Settlements Conditions->Employment Choices->Agricultural Land Transfer | 0.197 | 0.227 | 0.196 | 0.226 |
| Settlements Conditions and Employment Choices->Agricultural Land Transfer | 0.004 | 0.106 | 0.001 | 0.104 |
| Livelihood Capitals->Agricultural Land Transfer | 0.134 | 0.627 | 0.135 | 0.631 |
| Livelihood Capitals and Employment Choices->Agricultural Land Transfer | −0.467 | −0.036 | −0.467 | −0.036 |
| Livelihood Capitals->Employment Choices->Agricultural Land Transfer | −0.322 | −0.275 | −0.313 | −0.259 |
| Settlements Conditions, Livelihood Capitals, and Employment Choices->Agricultural Land Transfer | −0.229 | −0.103 | −0.238 | −0.111 |

## Appendix C

**Table A5.** Direct, indirect, and total effects of livelihood capitals and human settlements on agricultural land transfer.

| Model | ΔCMIN | ΔDF | $p$ |
|---|---|---|---|
| Measurement weights | 5.181 | 6 | 0.520817106 |
| Structural covariances | 18.799 | 16 | 0.279224094 |
| Measurement residuals | 38.882 | 26 | 0.050034415 |

**Table A6.** Direct, indirect, and total effects of livelihood capitals and human settlements on agricultural land transfer.

| Model | ΔCMIN | ΔDF | $p$ |
|---|---|---|---|
| Measurement weights | 67.094 | 20 | $5.391 \times 10^{-7}$ |
| Structural weights | 127.887 | 30 | $4.736 \times 10^{-14}$ |
| Structural covariate | 134.267 | 36 | $2.987 \times 10^{-13}$ |
| Structural residuals | 134.267 | 36 | $2.987 \times 10^{-13}$ |
| Measurement residuals | 498.861 | 64 | $1.323 \times 10^{-68}$ |
| Measurement weights | 67.094 | 20 | $5.391 \times 10^{-7}$ |

**Table A7.** Standardized bootstrap moderator–mediator test of the indifference region.

| Path | Eastern Region | | | | Central Region | | | | Western Region | | | |
|---|---|---|---|---|---|---|---|---|---|---|---|---|
| | Bias-Corrected 95% CI | | Percentile 95% CI | | Bias-Corrected 95% CI | | Percentile 95% CI | | Bias-Corrected 95% CI | | Percentile 95% CI | |
| | Lower | Upper | Lower | Upper | Lower | Upper | Lower | Upper | Lower | Upper | Lower | Upper |
| Settlements Conditions->Agricultural Land Transfer | 0.022 | 0.264 | 0.038 | 0.259 | 0.083 | 0.14 | 0.076 | 0.152 | 0.008 | 0.211 | 0.007 | 0.228 |
| Settlements Conditions->Employment Choices->Agricultural Land Transfer | 0.107 | 0.124 | 0.112 | 0.119 | 0.025 | 0.195 | 0.027 | 0.166 | −0.119 | −0.067 | −0.097 | −0.089 |
| Settlements Conditions and Employment Choices->Agricultural Land Transfer | 0.041 | 0.209 | 0.032 | 0.200 | 0.024 | 0.107 | 0.027 | 0.105 | −0.195 | −0.032 | −0.195 | −0.031 |
| Livelihood Capitals->Agricultural Land Transfer | 0.270 | 0. 566 | 0.233 | 0. 505 | 0.097 | 0. 381 | 0.102 | 0. 395 | 0.480 | 0.760 | 0. 451 | 0.744 |
| Livelihood Capitals and Employment Choices->Agricultural Land Transfer | −0.597 | −0.329 | −0.534 | −0. 303 | −0.634 | −0.381 | −0.626 | −0.375 | −0.755 | −0.500 | −0.711 | −0.485 |
| Livelihood Capitals->Employment Choices->Agricultural Land Transfer | −0.142 | −0.022 | −0.141 | −0.028 | −0.337 | −0.203 | −0.321 | −0.187 | −0.099 | −0.075 | −0.083 | −0.074 |
| Settlements Conditions, Livelihood Capitals, and Employment Choices->Agricultural Land Transfer | −0.264 | −0.077 | −0.263 | −0.075 | −0.320 | −0.135 | −0.314 | −0.132 | −0.204 | −0.012 | −0.209 | −0.016 |

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
