# Peer review of "The Causal Pathway of Rural Human Settlement, Livelihood Capital, and Agricultural Land Transfer Decision-Making: Is It Regional Consistency?"

_land, doi:10.3390/land11071077_

Round 1

Reviewer 1 Report

An important and current research problem was taken up in the manuscript submitted for review. All elements assessed in the scientific work have been prepared and presented at the highest level. Taking this into account, I recommend that you accept the manuscript for publication in Land, as submitted to the editor.

Author Response

Dear reviewer:

We greatly appreciate that the precious time you spent making comments and encouraging.

Reviewer 2 Report

The manuscript "The causal pathway of rural human settlement, livelihood capital, and agricultural land transfer decision-making: Is it regional consistency?" addresses a topic of great current interest. Research such as this is needed to learn more about the problems and difficulties experienced by populations in many rural settings, both in China and beyond. Thus, in circumstances of frequent economic and demographic declines in many rural areas, in which their populations are faced with the unfair positioning of rural values and the long-term isolation of urban-rural relations, migration to urban settings is often the main employment choice for the young generations.  Faced with this situation, it is necessary to seek ways of sustainable rural development that make rural territories attractive for populations and investment, and, therefore, that these territories keep their social and/or productive structures alive.

The authors construct a theoretical framework that, in their own words, "integrates rural human settlements (public resources) and household livelihood capitals (private assets) to explore the causal pathway of livelihood capitals and rural human settlements on an agricultural land transfer through rural employment." The manuscript is well written and reads easily. However, the authors should make the following suggested changes in order to make the objectives and achievements of their work stronger.

1) The abstract should be reworked to make it clearer for the readers the research objectives, the type of analysis that has been done and the findings attained.

2) The section "Materials and methods" is too brief.  The "Materials and methods" should be explained in more detail and, in addition, what is said in this section should be related with adequate reasonings to what has been said in the previous section devoted to the "Theoretical Framework".

3)The conclusions should be more extensive, systematically and clearly highlighting the findings and recommendations on how the institutions and persons responsible for implementing rural development public policies can put these findings into practice. In addition, in this last section it would be very appropriate for the authors to suggest some ideas on how the theoretical-methodological framework and the analytical process followed in this research could be extrapolated to similar studies from inside and outside China.

4) The authors should explain in more detail concepts such as "The Sustainable Livelihood Framework (SLF)" and especially what they mean by "household livelihood capitals". In my opinion, this concept is not sufficiently clear. Among other reasons, because it is said that "household livelihood capitals" are “private assets”, but however, in the sentence with quotation marks copied next, this concept is related to a series of dimensions, both public and private: "According to the SLF, an evaluation index system that measures livelihood capital can be constructed, which includes five dimensions (human-, nature-, physical-, financial-, and social- capital)".

5) In line 168 and following it says: "Grassroots government and village committees assist when households face livelihood difficulties. They can further create a safe living environment by mediating disputes, maintaining social order, and managing public affairs [53]." I have not found elsewhere in the manuscript references to "grassroots government and village committees" entities. In this regard, it should be noted that the role of the "grassroots government and village committees" is key in the processes of functioning, management and evolution of rural societies, as well as in achieving satisfactory results in the matter studied in the article. Therefore, I consider that it would be very appropriate for the authors to introduce in their manuscript some additional lines explaining how the said entities act and what functions they usually perform in Chinese rural societies.

6) Finally, the authors do not make clear what they mean by "social capital" either. The only sentence I have found that goes into a little more detail when talking about "social capital" is the one that begins on line 144 and following. It says: "Social capital denotes social networks and social resources that households rely upon to pursue sustainable livelihoods [17]. As the most important social organization in China, the village committee helps households to enhance their livelihood [39]. The networks between relatives and friends are the primary channels for households to obtain information and assistance. Additionally, social security is a redistribution of social resources, which is another important source of social capital [45]. Therefore, the village cadres, cash gifts, and social security are considered evaluation variables of social capital". However, given the high degree of polysemy of the expression "social capital", in any work in which this concept is discussed the first thing to do is to explain and make clear what in that work is understood by "social capital". In this respect, taking into consideration what Pierre Bourdieu said with respect to "social capital" could be very useful and orienting to the authors. In any case, whatever work the authors consult or may consult in this regard, the important thing is that they became convinced of how important it is to make clear to the reader what they mean by "social capital", so that one should avoid at all costs ambiguous and imprecise phrases such as the one in quotation marks above, in which it is certainly not at all clear what is meant by "social capital", but rather quite the opposite.

I hope that my previous comments, which have no other purpose than to make constructive criticisms and recommendations, will be of help to the authors and that they will be able to improve their manuscript.
